# Using search engine data to gauge public interest in mental health, politics and violence in the context of mass shootings

T. Vargas[1]*, J. Schiffman[2,3], P. H. Lam[1], A. Kim[1], V. A. Mittal[1,4,5,6,7]

**1** Department of Psychology, Northwestern University, Evanston, IL, United States of America,
**2** Department of Psychology, University of Maryland, Baltimore County, Baltimore, MD, United States of America, **3** Department of Psychological Science, University of California, Irvine, Irvine, CA, United States of America, **4** Department of Psychiatry, Northwestern University, Evanston, IL, United States of America, **5** Department of Medical Social Sciences, Northwestern University, Evanston, IL, United States of America, **6** Institute for Policy Research, Northwestern University, Evanston, IL, United States of America, **7** Institute for Innovations in Developmental Sciences, Northwestern University, Evanston, IL, United States of America

* teresavargas@u.northwestern.edu

**Data Availability Statement:** All data was extracted from publicly available google trends database: https://trends.google.com/trends/?geo=US.

## Abstract

Despite significant potential for providing insight to private perceptions and behaviors, search engine data has yet to be utilized as a means of gauging the U.S. public's interest and understanding of mental health in the context of gun violence and politics. An analysis of Google Trends revealed that Mental health searches increased in volume starting in the beginning of the current decade. Notably, both "mental health" and "gun(s)" were searched with greater frequency the week after the mass shooting events occurred. Related searches after the event also observed a significant increase in interest in mental health and gun regulation, legal reform, mass shootings, and gun(s). Results suggest that the American public's perception of mental illness increasingly incorporates associations with themes of violence and politics, which becomes more apparent surrounding mass shooting events. Future studies are needed to determine implications for stigmatization of vulnerable groups, and possible relations to media coverage.

## 1. Introduction

The media has increased its coverage of mental health in recent years [1]. Concurrently, the United States has seen a stark increase in gun violence events [2–4]. Media coverage has often cited mental health as a cause for mass shooting events specifically [1, 5]. While this has been occurring, it is unclear if public perceptions of mental health are also changing. While common public polling methods have provided some insight [6–8], the approach can be vulnerable to factors such as desirability and recall bias. The current investigation used search engine data (Google Trends) to gauge public attention to mental health, violence, and gun violence events. The method of choice offers an estimate of private attention to a subject. The relative anonymity of internet search engines provides data less tainted by how individuals would like to be perceived. It was of interest to determine whether volume of searches for mental health have

**Funding:** The research reported in this manuscript was supported by the National Institute Of Mental Health of the National Institutes of Health under Award Number F31MH119776 (T.V.). The content is solely the responsibility of the authors and does not necessarily represent the official views of the National Institutes of Health. This work was also supported by grants R01MH112545, R21/R33MH103231 and R21MH110374 (V.A.M).

**Competing interests:** The authors have declared that no competing interests exist.

shifted over the years, and whether related search terms concerned subjects surrounding violence, gun violence, politics/government and mental health. In addition, the study examined immediate shifts in perception following mass shooting events over recent years. Specifically, we evaluated whether searches related to guns and mental health changed the week after a mass shooting event, and whether the volume of these searches increased after the event. In this way, the investigation provides a perspective on individuals' private behavior, and possible effects that violence and gun politics may have on the manner that mental health is perceived on a population level.

Mental health is a critical public health concern, with 1 in 5 adults in the U.S. experiencing mental illness in a given year [9], and lifetime prevalence estimated at up to 1 in 2 Americans [10]. While there is evidence to suggest more individuals are seeking treatment for mental illness today than in past years, a gap remains between individuals in need of treatment and those who ultimate pursue it [6, 8, 10–13]. One key barrier to treatment-seeking is stigma surrounding mental health, at the individual and societal level [7, 14–16]. On a broader societal level, public perceptions of mental health could shape policy efforts surrounding mental illness [8]. Thus, it is of utmost necessity to investigate factors such as public perception that may contribute to barriers for mental health treatment. What is more, there is evidence to suggest media coverage of issues surrounding mental health has increased over the years [1, 5]. While this may be the case, it is unclear exactly how media coverage of mental health affects public interest on the issue. Further, it is also unclear what this interest in mental health is in regards to (e.g. treatment seeking, violent events, advocacy, or something else?). Understanding these factors would aid in clarifying public and private barriers to mental illness prevention, intervention, and treatment efforts.

There is reason to hypothesize the national discourse on mental health has shifted over the years. A recent investigation looked toward media portrayals of mental illness to gain a greater understanding of sources possibly impacting public perceptions. Using random sampling to extract 400 news stories from 1995 to 2014, McGinty and colleagues found that in the latter decade, stories were more likely to mention mass shootings by individuals with mental illness [5]. The most frequently occurring topic of coverage across the study period was violence, which was mentioned 55% of the time. In contrast, only 14% of news stories depicted successful treatment or recovery from mental illness. Strikingly, there is strong evidence that media portrayals of topics related to mental health are likely to have an impact on individual attitudes toward mental illness [6–8, 17–21]. For example, studies found that increased depictions of untreated individuals diagnosed with mental illness can result in increased social distance from groups suffering from mental illness; this, in turn, has been found to lead to increased isolation of these individuals [15, 19, 22, 23]. Furthermore, stigmatizing attitudes toward mental illness have been linked to reduced support for policy supporting mental health [7]. Understanding how public perception is impacted on an individual basis could thus be critical for gauging the societal effects of media coverage of mental health concerns. In the case of violence, it could be particularly impactful, given evidence that the link between violence and mental illness is reliable but often overestimated in magnitude [24], with population-wide studies having shown that only around 4% of violence in the U.S. can be attributed to individuals diagnosed with a psychiatric disorder [25].

Google Trends (https://trends.google.com/trends/?geo=US) provides a unique and valuable big data approach to examining public interest and perceptions [26]. Search engine data is becoming increasingly pertinent, especially as the proportion of individuals using the Internet in the U.S. is estimated as high as 90% in recent years [27]. Moreover, Google is estimated to capture 90% of desktop search engine searches [28], and 95% of mobile device searches [29]. The situations under which individuals perform Google searches provide a certain degree of

anonymity. Searching online, often alone, not as part of official surveys or experiments lends itself to a reduction in social desirability bias [26, 30]. Capturing searches as they occur in real time rather than asking participants to recall their interest over time also removes confounds related to recall bias. Google Trends has been increasingly used to predict influenza epidemics [31], determine the effect of race on presidential polling [30], and even quantify interest in birth control and abortion [32]. Thus, it may lend itself particularly well to gathering public opinion and interest on topics that would otherwise be sensitive and difficult to estimate [33–35]. In addition, Google Trends offers indices of keyword search popularity, along with noting search topics that often accompany keywords of interest. This affords us the unique opportunity not only of gauging how frequent a certain search term is, but also of understanding the context in which the search is being conducted. Thus, Google Trends affords a necessary lens into private searching behavior at the aggregate population level, while also allowing us to understand the context and implications of certain searches. And yet, this approach has yet to be used in the context of questions regarding mental health, violence, and politics.

The present study first aimed to visualize public interest in mental health over time through Google Trends search data. Scaled scores were extracted from January 2004 through the end of July 2019. It was also of interest to determine whether searches related to mental health increased for certain subject domains (Law, Politics/Government, Crime/Violence, Mass Shootings, Stigma, Gun(s)) over time. Then, we honed our focus on specific time frames surrounding gun violence events, examining whether searches for words such as "Gun(s)" and "Mental health" increased the week after these gun violence events. Finally, we analyzed how related searches for these key words (i.e. "gun(s)" and "mental health") and relevant subject domains (Law, Politics/Government, Crime/Violence, Mass Shootings, Stigma, Gun(s), Mental health) manifested the week before versus after the events.

## 2. Methods

### 2.1. Search data

Google Trends provides search volume data by compiling a score which consists of relative frequency of searches of specific keywords at the time and location specified [36]. The data is publicly available (https://trends.google.com/trends/?geo=US). The content in this study is solely the responsibility of the authors and does not represent the official views of Google. Scores range from 0–100; a score of 100 represents the greatest proportion of searches for that keyword as a function of total searches at the specified time and place (i.e. the most "popular" search term for that specified timespan and location). A value of 50 denotes a keyword that is half as popular. Thus, these scores provide a measure of relative frequency of searches, compared to total searches for all key words at a specific date and location. For example, in the case of searches of the term "mental health" during March 2014, the scaled score represents the following value:

$$\frac{Number \; of \; searches \; including \; mental \; health \; in \; the \; U.S. \; in \; March \; 2014}{Total \; number \; of \; Google \; searches \; in \; the \; U.S. \; in \; March \; 2014}$$

### 2.2. Mental health searches over time

**2.2a Extraction of "Mental Health" data searches over time.**   Scaled scores (described above) were extracted for months from January 2004 to July 2019, as well as graphed to observe possible trends in the U.S. over the years. Data was extracted during the end of July of 2019. Each data point represents one month within this time frame (Fig 1).

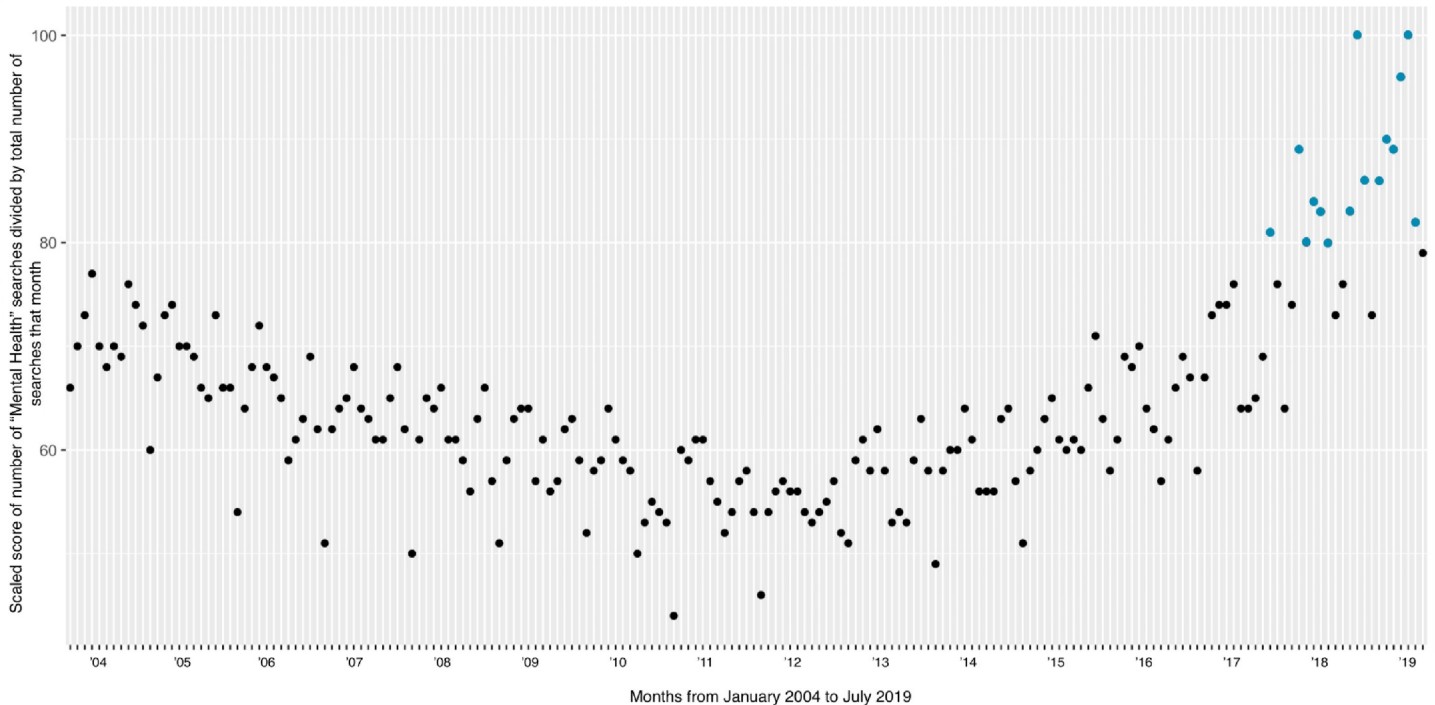

**Fig 1. "Mental Health" searches over time, from 2004–2019.** The highest 15 peak proportions of "Mental Health" searches are highlighted in blue.

**2.2b Descriptive information of related searches for "Mental Health" terms over time.**
Given the observed trend toward greater proportion of mental health searches over the years, the study examined topics of these searches, and whether the search topics would shift over time. Since the aim was to understand the increasing volume of searches over the years, it was of particular interest to determine topics of searches when the proportion of searches for mental health was highest. Thus, the month within each year with the highest scaled score was chosen for describing related terms searched alongside "Mental Health" for yearly peaks of "Mental Health" searches. In the case that multiple months in the same year had the same scaled score, the outputs for all of these months were aggregated into the yearly peak. Related entities/topics were extracted for the U.S. These denote topics that were most often searched along with the term of interest (i.e. mental health). For example, if a related search term is "treatment centers," this means that the term was searched alongside this topic (i.e. "mental health treatment centers"). Extraction of related entities provides both "Top" and "Rising" Google sections. The "Top" section denotes entities/topics that are often searched along with the key term of interest. The "Rising" section, in contrast, denotes entities/topics that have recently risen in popularity, being more often searched alongside the key term of interest. For each yearly peak, "Top" and "Rising" related searches were compiled into a single list of related terms. In the case that related terms were repeated in both the "Rising" and "Top" section (i.e. in the case that the term was both popularly searched along with the key term, and had simultaneously recently had a spike in popularity), these terms were removed so that each term came up once per extracted yearly peak month. Related search topics per yearly peak were sorted into categories by 2 independent raters who converged on ratings with kappas > 0.80. Related searches consisting of topics of interest for the current investigation included searches related to Crime/Violence, Gun Regulation, Guns and Gun parts, Law, Mass Shootings, Politics and Government, and Stigma. Topics of interest to the study questions searched alongside

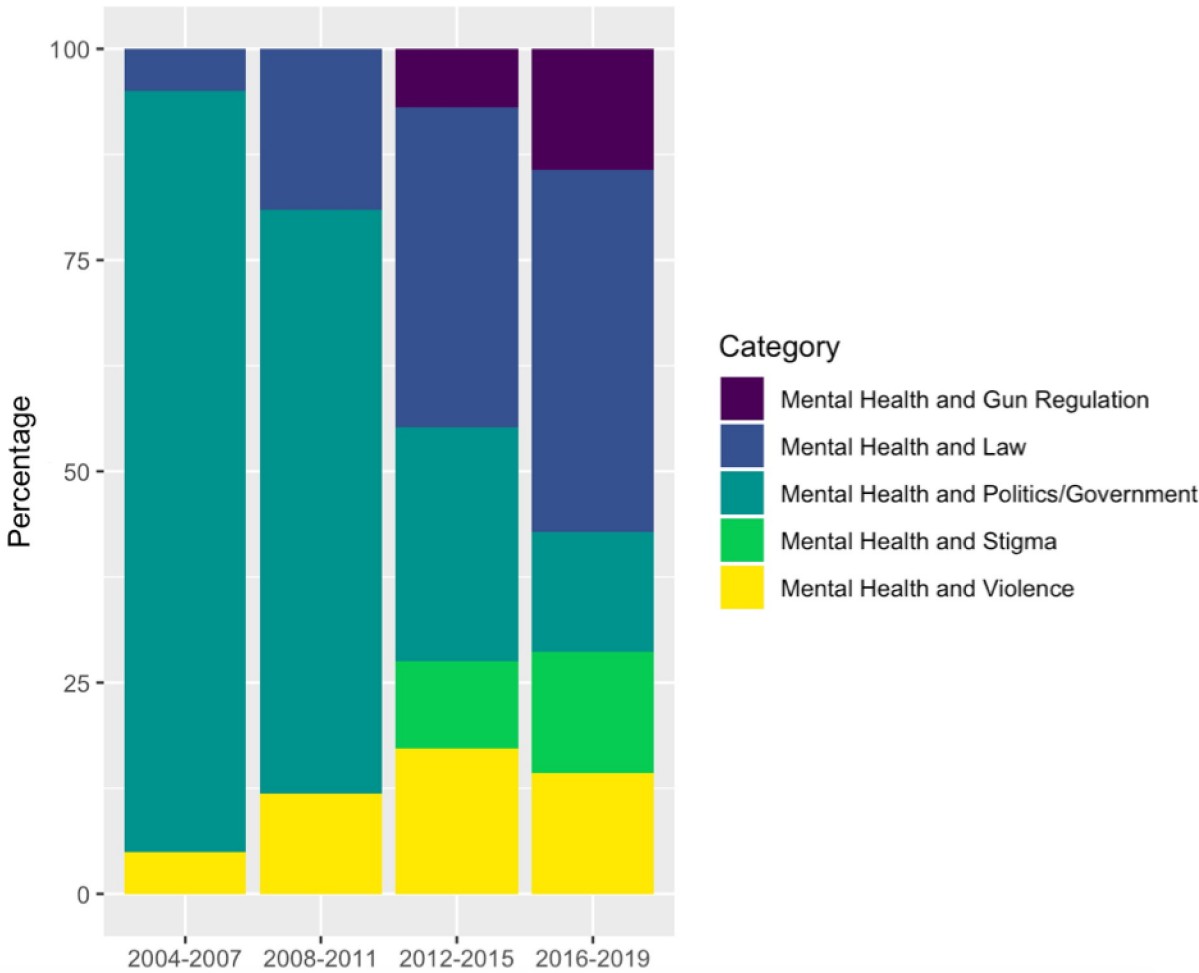

**Fig 2. "Mental Health" related searches in yearly peaks from 2004 to 2019, specifically presenting entities regarding Gun Regulation, Law, Politics/Government, Stigma, and Violence.**

the "Mental Health" key term were presented in 4-year periods: 2004–2007, 2008–2011, 2012–2015, and 2016 to 2019 (Fig 2).

### 2.3. Searches before and after mass shooting events

**2.3a. Inclusion strategy for mass shootings.** The Gun Violence Archive (GVA) is a non-profit corporation that publishes data on gun violence in the United States [37]. Data on mass shootings dating back to 2014 was extracted from this organization on July 2019. Included events (1) were designated as "mass shootings" by the media (as determined by internet searches of media coverage) and (2) had 10 or more people injured by firearm. Included incidents meeting these criteria totaled 15 cases (Table 1).

**2.3b. Selection of key search terms.** Search terms of interest included "Mental Health" as well as "Gun(s)" terms. "Mental Health" was chosen as an umbrella term in order to broaden the scope of searches. "Gun(s)" was additionally selected as an umbrella term in order to capture subjects related to violence and gun control, as well as mental health, among other topics of interest. "Gun(s)" was additionally chosen as a term in order to verify whether searches relating to mental health and violence/guns would exhibit a bidirectional pattern. For example,

**Table 1. Included mass shootings from 2014–2019 along with shooting characteristics.**

| # | Incident Date | State | # fatalities/# injured |
|---|---|---|---|
| 1 | 1-Oct-15 | Oregon | 10/9 |
| 2 | 2-Dec-15 | California | 16/19 |
| 3 | 12-Jun-16 | Florida | 50/53 |
| 4 | 7-Jul-16 | Texas | 5/9 |
| 5 | 6-Jan-17 | Florida | 5/6 |
| 6 | 10-Sep-17 | Texas | 9/1 |
| 7 | 1-Oct-17 | Nevada | 59/441 |
| 8 | 5-Nov-17 | Texas | 27/20 |
| 9 | 14-Nov-17 | California | 6/12 |
| 10 | 14-Feb-18 | Florida | 17/17 |
| 11 | 18-May-18 | Texas | 10/13 |
| 12 | 27-Oct-18 | Pennsylvania | 11/7 |
| 13 | 7-Nov-18 | California | 13/2 |
| 14 | 15-Feb-19 | Illinois | 6/6 |
| 15 | 31-May-19 | Virginia | 13/6 |

[a]Reasons cited by the media upon conducting a search.

[b]MI = mental illness. The column describes whether the incident or perpetrator received coverage in the media related to the perpetrator's mental illness.

[c]As measured by the Pew Research Center [38].

whether those searching for "Gun(s)" would often search for terms related to mental health in the same manner that those searching for "Mental Health" would often search with relation to gun(s)/violence.

**2.3c. Main effect analyses.** Search terms were extracted for the week before the 15 events, and the week after the 15 events, for each of the 50 states and Washington D.C. The data extraction took place in July 2019. Before and after scores of relative search volume were extracted for key terms ("Mental Health" and "Gun(s)"). A fixed effects panel regression using the Least Squares Dummy Variable (LSDV) approach was then conducted [39]. Searches from each state the week before and after the events were compared, with the 15 events accounted for as fixed effects. An interaction of state where the event occurred by timepoint was tested to see whether change in search volumes in the state the event occurred would be significantly different from change in search volumes in states where the event did not occur.

**2.3d. Descriptive data of related search terms.** Related topics searched along with our key words of interest were extracted for the week before and the week after the 15 events. The data extraction took place in July 2019. Related search terms denote search terms that were most often searched alongside our selected search terms (i.e. "Mental Health" and "Gun(s)"). The "Top" and "Rising" sections of related entities/topics were compiled into one list. In the case that an entity/topic was repeated in the "Rising" or "Top" list, the entity/topic was only counted once. Categorization was completed using the same procedure detailed earlier. Total related searches were presented divided by sub-categories to denote proportions for different topics of interest. In addition, the breakdown of related search terms of interest (i.e. topics often searched alongside our key words) was presented for before and after the events. Related searches comprising topics of interest included Crime/Violence, Gun Regulation, Guns and Gun parts, Law, Mass Shootings, Politics and Government, Stigma and Mental Disorders. Topics of interest were decided based on our theoretical interest in mental health and topics related to mass shootings (for example, since "toys" is not theoretically related to mass

shooting events, this category was not deemed a topic of interest). For searches related to
"Mental Health," "Mental Disorders" was not treated as a topic of interest to avoid redundancy
(since the search term concerns mental health, related searches in the category of "mental dis-
orders" will be vastly overlapping with the search term itself, thus not providing any additional
meaningful information). Likewise, in the case of related searches for the "Gun(s)" keyword,
"Guns and Gun parts" was not treated as a topic of interest.

 **2.3e. Related search terms for the week before and week after.** Related searches for
"Gun(s)" and "Mental Health" key words were extracted for the week before and the week
after the 15 events. The week before was chosen as a comparison point prior to the event taking
place. Each related search was dummy coded as "topic of interest" or "not topic of interest."
Percentage of topics of interest among total related searches were computed for the week
before and after (Figs 3 and 4). A fixed effects panel regression using the LSDV approach (to
account for events as fixed effects) was conducted to determine whether proportion of related
search terms would increase after the event occurred.

# 3. Results

## 3.1. "Mental Health" searches from 2004–2019

The top 15 monthly "peaks" with the highest proportion of searches occur in recent years (Fig
1). Beginning around 2012, the proportion of searches for "Mental Health" relative to total
monthly searches steadily increases over the months, peaking in the last 2 years.

## 3.2. "Mental Health" related searches for yearly peaks from 2004–2019

Over the years, the proportion of searches related to "Mental Health and Law" increases (from
5% in 2004–2007, to 43% of related searches in 2016–2019). The proportion of searches related
to "Mental Health and Gun Regulation", and "Mental Health and Stigma" also increased; both
of these terms were not in top/rising related searches from 2004–2007, meaning they had not
recently risen in popularity and did not constitute searches that were frequently searched

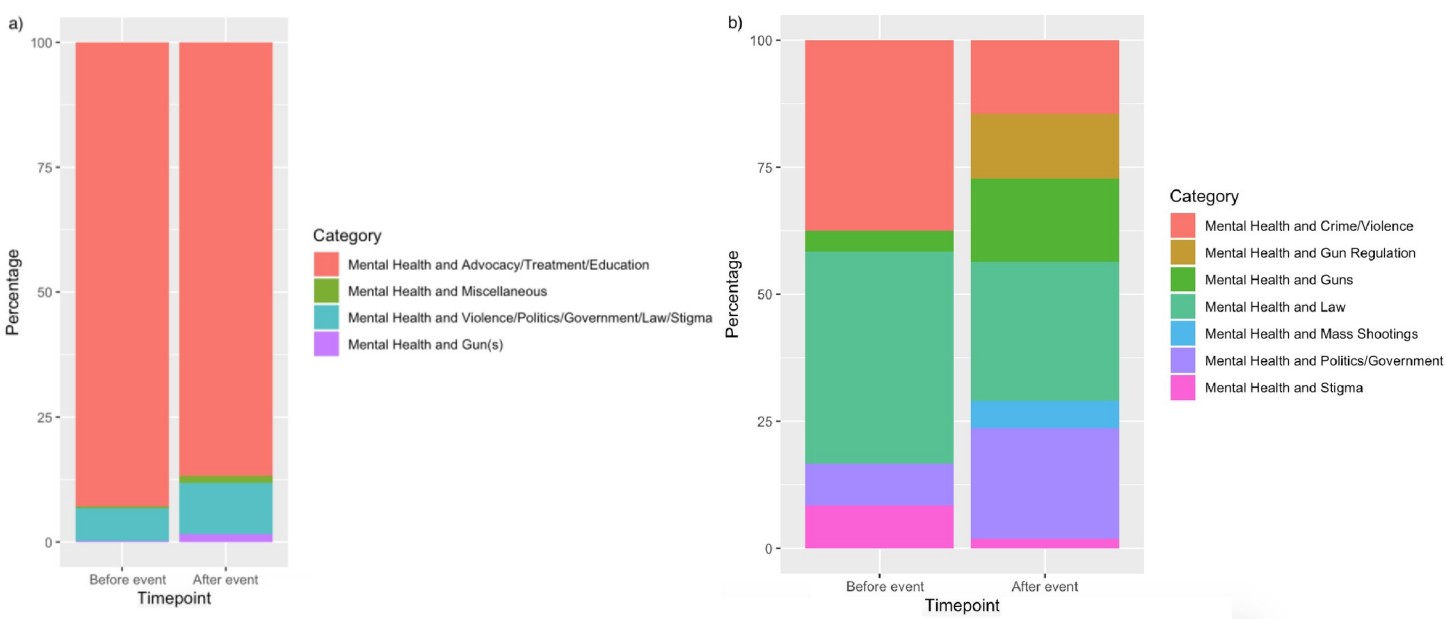

**Fig 3.** Mental health key word related searches, categorized for overall keyword related searches (a) and broken down to specify relative percentages of categories among
topics determined to be event-related (b).

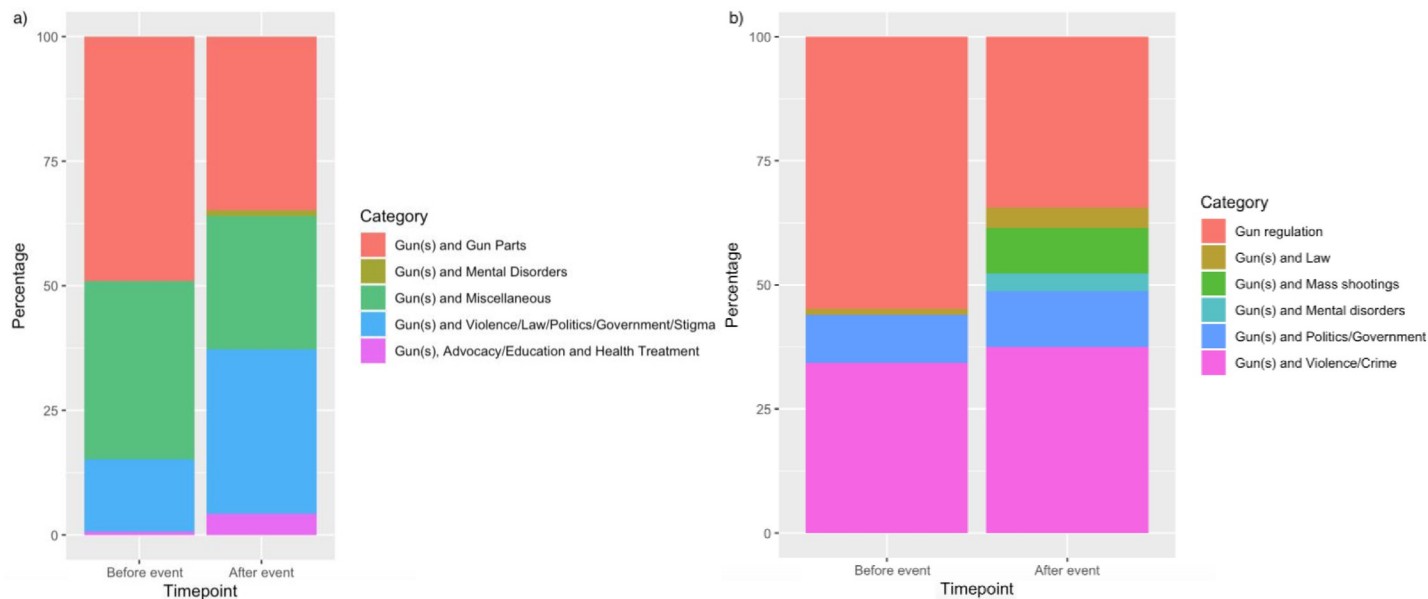

**Fig 4.** Gun/guns key words related searches, categorized for overall keyword related searches (a) and broken down to specify relative percentages of categories among topics determined to be event-related (b).

alongside our term of interest. However, by 2016–2019, "Mental Health and Gun Regulation" and "Mental Health and Stigma" went on to constitute 14% of top/rising searches of interest in 2016 to 2019; Fig 2.

### 3.3. Keyword searches before and after mass shooting events

For the "mental health" key term, the model was significant, $R = 0.30$, $R^2 = 0.09$, $F = 0.892$, $p < 0.001$. Relative to the week before the event, there was a trend toward larger volume of "mental health" searches, $b = 1.47$, $SE = 0.82$, $t = 1.80$, $p = 0.07$. A main effect was not observed for whether the search occurred in the state the event occurred, $b = -13.25$, $SE = 9.20$, $t = -1.44$, $p = 0.15$. The interaction of event timepoint and search occurring in-state/out-of-state was not significant, $b = 0.80$, $SE = 5.82$, $t = 0.14$, $p = 0.89$.

The model of searches for the "gun" key term was likewise significant, $R = 0.35$, $R^2 = 0.12$, $F = 12.04$, $p < 0.001$. Relative to the week before the event, a larger volume of "gun" searches was observed the week after, $b = 5.42$, $SE = 0.64$, $t = 8.45$, $p < 0.001$. A significant main effect was not observed for whether the search occurred in the state the event occurred ($b = -10.96$, $SE = 7.26$, $t = -1.51$, $p = 0.13$), and neither was an interaction ($b = 6.97$, $SE = 4.59$, $t = 1.52$, $p = 0.13$). As expected, the model for searches for the "guns" key term was also significant, $R = 0.33$, $R^2 = 0.11$, $F = 10.70$, $p < 0.001$. Compared to the week before events occurred, a larger volume of "guns" searches was observed the following week, $b = 4.80$, $SE = 0.79$, $t = 6.05$, $p < 0.001$. As was the case for other key terms, a significant main effect was not observed for the search occurring in the state where the event occurred ($b = -15.44$, $SE = 8.95$, $t = -1.72$, $p = 0.09$), nor was a significant interaction observed ($b = 7.07$, $SE = 5.66$, $t = 1.25$, $p = 0.21$).

### 3.4. Descriptive information on related searches alongside keywords before and after mass shooting events

As expected, an increase in the proportion of related searches (in categories of interest) the week after the event was observed for mental health and gun(s) related searches (Figs 3 and 4).

For "Mental Health" total related searches, there were observed increases in proportions for "Mental Health and Gun(s)," and "Mental Health and Violence/Politics/Government/Law/Stigma" (Fig 3A). When honing into topics of interest to the current investigation, a greater proportion of searches can be observed for "Mental Health and Gun Regulation," "Mental Health and Guns," and "Mental Health and Mass Shootings" the week after the events (Fig 3B). For example, both "Mental Health and Mass Shootings" and "Mental Health and Gun Regulation" went from not appearing in top/rising searches, to comprising 18% of top/rising searches of interest after the event. The proportion of searches for "Mental Health and Politics/Government" also increased from 8% of searches of interest the week before to 22% of searches of interest the week after. For Gun(s) total related searches, in turn, there were observed increases in proportions for "Gun(s) and Mental Disorders," "Gun(s) and Violence/Law/Politics/Government," and "Gun(s), Advocacy/Education/ and Health Treatment" (Fig 4A). When observing topics of interest for the present study, there were observed increases in proportion the week after for searches related to "Gun(s) and Law," "Gun(s) and Mass Shootings," "Gun(s) and Mental Disorders," and "Gun(s) and Politics/Government" (Fig 4B). Searches for "Gun(s) and Mass Shootings," for example, went from not appearing in top/rising related searches, to comprising 9% of top/rising related searches.

### 3.5. Related search terms for mass shooting events

For the "mental health" search term, across the 15 events for the week before, there were 404 total related search terms. For the week after, there were 413 related search terms. For the week before the events, 8.17% of related search terms were topics of interest. For the week after, 14.29% of related searches were topics of interest (Fig 3). For the "gun(s)" search term, across the 15 events for the week before, there were 921 total related search terms. For the week after, there were 725 related search terms. For the week before the events, 16.40% of related searches were topics of interest. For the week after the events, 48.69% of related searches were topics of interest (Fig 4). The model of related searches before and after the event for the "mental health" key term was significant, $R = 0.25$, $R^2 = 0.06$, $F = 4.04$, $p < 0.001$ —related searches that pertained to law, politics/government, crime/violence, mass shootings, stigma, and guns significantly increased in proportion the week after the event, $\beta = 0.08$, $t = 2.44$, $p = 0.015$. Likewise, the model of related searches for "gun(s)" key terms was significant, $R = 0.28$, $R^2 = 0.08$, $F = 12.39$, $p < 0.001$. The "gun(s)" related searches that pertained to law, politics/government, crime/violence, mass shootings, stigma, and mental health significantly increased in proportion the week after the event, $\beta = 0.22$, $t = 10.35$, $p < 0.001$.

## 4. Discussion

The present investigation explored interest in mental health over time, as well as public perception of mental health in relation to mass shootings, violence, and politics/government. Over the years, and especially starting at the beginning of the decade, the proportion of mental health searches was observed to increase—all 15 peak months occurred within the past two years. These patterns suggest an increased interest in mental health in the United States in recent years. Further, related searches for yearly peaks were examined in order to better understand in what context mental health was being searched. Over the years, searches for mental health in relation to politics/government and gun regulation increased in proportion to other topics of interest. These observations support the notion that the general public is increasingly associating mental health to these concerns. Furthermore, the study aimed to focus on mass shooting events specifically, in order to determine whether searches for key terms and related topics of interest increased immediately following mass shooting events. An increase in

searches for both "mental health" and "gun(s)" was observed the week after the mass shooting events. In addition, related searches for topics of interest (i.e. mental health and gun regulation, politics/government, mass shootings, and guns) increased the week following mass shooting events. Taken together, results support the notion that the discourse surrounding gun politics/government and gun regulation increasingly includes mental illness as a causal factor. In addition, findings suggest that individuals in the U.S. have heightened interest in this association, to the extent that they are increasingly seeking answers on these subjects after shooting events. Observed patterns have poignant ramifications for how vulnerable populations with mental illness are perceived and conceptualized, which in turn has the potential to impact treatment, prevention and intervention efforts as well as public policy.

Examining monthly search volumes for "mental health" revealed a striking trend: since the beginning of the decade, searches for "mental health" have increased in volume. There are multiple interpretations for this observation. There is evidence that a greater percentage of U. S. adolescents and young adults in the late 2010s (relative to the mid-2000s) are experiencing certain kinds of mental health disorders [40]. This seems to be the case for both self-report of symptoms and other sources (such as hospital records for suicide ideation and attempts) [40, 41]. Supporting this notion, a majority of related overall searches pertained to mental health treatment, advocacy and education. Perhaps this, combined with adolescent and young adult's increased internet usage compared to earlier generations, is a contributing factor [42]. Another possibility is that mental health awareness over the years has increased, with amplified media attention on mental health issues occurring in tandem with increases in public interest. While evidence of increased media attention to mental health coverage has been found in the U.K. [1, 43], evidence for the U.S. has been mixed. Indeed, one study found that in the U.S., print and television news coverage of mental illness trended downward, rather than upward, from 1995 through 2014 [5]. However, these stories covering mental health in the U.S. were also found to be more likely to make the "front page" in later years, suggesting increased prominence of the subject relative to other covered topics in the U.S. The same investigation found that in the U. S., mental health media coverage most frequently related to violence. Media coverage of mental illness relating to gun violence and mass shootings has also been pervasively reported [44]. Related search data for yearly peaks of "mental health" searches over the years in the present investigation observed the proportion of related searches for "Mental Health and Gun Regulation" and "Mental Health and Law" to increase over recent years. Thus, another candidate contributor to increased public interest in mental health (as measured by internet searches) is mass shootings and gun violence more broadly.

Exploring this notion further, the study examined search data before and after major mass shootings events in the U.S. As expected, searches for "gun(s)" significantly increased the week after the mass shooting events, and "mental health" showed a strong trend toward increasing the following week, suggesting a heightened salience and interest in these subjects. Moreover, examining related searches for "mental health" and "gun(s)" key words allowed us a unique perspective for understanding the context in which these searches were taking place. For mental health searches, individuals searched in significantly greater proportions the week after the events with regards to mental health and gun regulations, guns, mass shootings, and politics/government. For gun(s) searches, individuals searched in significantly greater proportions the week after the events in terms of guns and mass shootings, mental disorders, and politics/government. Current findings suggest that public perception associates mass shootings/gun violence to mental illness to a certain degree, and that this becomes especially evident soon after mass shooting events occur. Significant differences in search volume were not found in the states where the shooting events occurred (relative to states in which the event did not occur);

future investigations could instill more fine-grained approaches to inform the manners in which geography could interact with search volumes.

The observation that individuals' private understanding of mental illness is increasingly linked to gun violence and politics/government has several potentially troubling implications. First, a strong link between violence and mental illness seems to be lacking robust evidence [24]. Population-wide studies have shown that only around 4% of violence in the U.S. can be attributed to individuals diagnosed with a psychiatric disorder [25]. Most common mental disorders are not correlated with violence, and the link between serious mental illness and violent acts is in large part attributable to substance abuse [24, 44–50]. Moreover, mental health being increasingly linked to violence in the minds of Americans could facilitate greater stigmatization of individuals with mental illness [7, 14–16]. Experimental studies have shown that negative attitudes toward individuals with serious mental illness increase following coverage of mass shootings where mental health is cited as a possible cause [6]. Importantly, this was not the case with mass shooting stories that covered gun restriction policy without mentioning mental illness.

Increased stigmatization of individuals with mental illness has been linked to a host of adverse functional outcomes, and stands as a primary barrier for treatment seeking in individuals diagnosed with a psychiatric disorder [14, 17, 51, 52]. On a societal level, relating gun violence and violence more broadly to mental illness could exert pressure on individuals with mental illness not to seek treatment so as to not be perceived as "dangerous," or for fear of learning (as a result of internalized stigmatization), for example, that they may in fact be violent upon seeking help. Taken a step further, widespread stigmatization of mental illness could lead to threats to the civil liberties of individuals with mental illness (e.g. external monitoring, profiling, involuntary commitment). There is also evidence that stigmatization has an impact on a broader policy level. For example, stigma toward individuals with mental illness has been related to lower support for policies to expand insurance and funding for mental health treatment [7]. Present study results suggest that to a certain degree, widespread media coverage of mass shooting events could be occurring alongside increased salience in public perception of relations between violence, politics/government and mental health. This, in turn, could lead to stigmatization of an already vulnerable group, which in turn could result in shifts in policy support at the societal level, and barriers to treatment seeking on an individual level. Future efforts geared toward increasing public education and advocacy of mental illness will be critical toward managing some of these challenges.

The current investigation provided a glimpse into private perceptions of mental illness following mass shooting events, in addition to elucidating patterns in volumes of searches related to mental health over the decade. Studies gathering how amount, duration, and prominence of media coverage relates to public attitudes and stigma will be a necessary future step. Mental illness is a pressing public health concern, given the high prevalence and gap in treatment of mental disorders [11, 13]. It will therefore be critical to examine factors to alleviate the possible burden of harmful media coverage, stigma and barriers to treatment in this vulnerable population. While the present investigation focused on nationwide internet searches in the US, it will be informative for future investigations to explore demographically distinct subgroups, to see whether differing search patterns emerge. In addition, it would be ideal to combine data on search engine searches with self-report and population surveys, in order to gauge sources of discrepancy and convergence. These questions will gain more relevance, and ultimately gain the potential to inform public health policy and efforts as this line of study progresses. Although our preliminary results suggest that greater numbers of searches after mass shootings, it is possible that in some circles inroads have been made to reduce stigma around mental health and that stigma reduction efforts have contributed to increased Google searching in an

effort for people to learn more. More work at the individual level to understand what drives these searches would add an additional level of context and depth to this field of inquiry.

## Author Contributions

**Conceptualization:** J. Schiffman, V. A. Mittal.

**Data curation:** A. Kim.

**Formal analysis:** T. Vargas, J. Schiffman, P. H. Lam, A. Kim, V. A. Mittal.

**Methodology:** T. Vargas, J. Schiffman, A. Kim, V. A. Mittal.

**Resources:** V. A. Mittal.

**Supervision:** V. A. Mittal.

**Visualization:** V. A. Mittal.

**Writing – original draft:** T. Vargas, J. Schiffman, V. A. Mittal.

**Writing – review & editing:** P. H. Lam.

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
