## [Decision Letter · Decision Letter 0]

11 May 2020

PONE-D-20-02370

Using search engine data to gauge public interest in mental health, politics and violence in the context of mass shootings

PLOS ONE

Dear Ms. Vargas,

I write you in regards to the manuscript PONE-D-20-02370 entitled “Using search engine data to gauge public interest in mental health, politics and violence in the context of mass shootings” which you submitted to PLOS ONE.

I have solicited advice from two expert Reviewers, who have returned the reports shown below. Both Reviewers are generally positive about the paper and recommend “Major Revisions”. However, they also raise a number of important issues that need to be resolved before the paper can be publishable. For example, the two Reviewers recommend to adopt a more sophisticated statistical approach. In this respect, they offer different suggestions that could improve the results of the paper.

Based on the Reviewers' reports and my own reading of the paper, I came to the decision to offer you the opportunity to revise the manuscript. If you decide to prepare a substantially revised version of the paper, please provide a detailed response to both Reviewers regarding how you have addressed their concerns. If you resubmit, I would ask the same two Reviewers to review again the paper and I would need to see substantial improvements in their assessments in order to proceed with the paper after the next round.

We would appreciate receiving your revised manuscript by Jun 18 2020 11:59PM. To enhance the reproducibility of your results, we recommend that if applicable you deposit your laboratory protocols in protocols.io, where a protocol can be assigned its own identifier (DOI) such that it can be cited independently in the future. For instructions see: http://journals.plos.org/plosone/s/submission-guidelines#loc-laboratory-protocols

We look forward to receiving your revised manuscript.

Kind regards,

Luis M. Miller, Ph.D.

Academic Editor

PLOS ONE

Journal Requirements:

2. In your Methods section, please include additional information about your dataset and ensure that you have included a statement specifying whether the collection method complied with the terms and conditions for the websites from which you have collected data.

Reviewers' comments:

Reviewer's Responses to Questions

**Comments to the Author**

1. Is the manuscript technically sound, and do the data support the conclusions?

Reviewer #1: Partly

Reviewer #2: Partly

2. Has the statistical analysis been performed appropriately and rigorously? 

Reviewer #1: Yes

Reviewer #2: Yes

3. Have the authors made all data underlying the findings in their manuscript fully available?

Reviewer #1: Yes

Reviewer #2: Yes

4. Is the manuscript presented in an intelligible fashion and written in standard English?

Reviewer #1: Yes

Reviewer #2: Yes

5. Review Comments to the Author

Reviewer #1: This paper uses Google Trends to assess the impact on mass shooting events on searches for guns and mental health. Overall, this paper is an exploratory project with few outright hypotheses or hypotheses tests. I think considering this would help this paper and would extract more out of Google Trends.

There are several pieces of research that highlight the utility and validity of Google Trends missing from this piece. Some important ones are:

Mellon, Jonathan. 2013. “Where and When Can We Use Google Trends to Measure Issue Salience?” PS: Political Science & Politics 46 (2): 280–90. https://doi.org/10.1017/S1049096513000279.

———. 2014. “Internet Search Data and Issue Salience: The Properties of Google Trends as a Measure of Issue Salience.” Journal of Elections, Public Opinion and Parties 24 (1): 45–72. https://doi.org/10.1080/17457289.2013.846346.

Reilly, Shauna, Sean Richey, and J. Benjamin Taylor. 2012. “Using Google Search Data for State Politics Research An Empirical Validity Test Using Roll-Off Data.” State Politics & Policy Quarterly 12 (2): 146–59. https://doi.org/10.1177/1532440012438889.

A crucial take-away from these papers is that, when using Google Trends, the most appropriate model is a cross-sectional time series. In the case of this paper, the independent variable should be a fixed effect for the gun violence event which allows for a clean pre-post change estimation. As it stands now, this paper is descriptive—which is fine—but the use of Google Trends allows for so much deeper analysis.

As for the mental health aspects, again, as this is merely descriptive with no large-scale content analysis for media agenda setting/framing/priming, I am not yet persuaded. I am just not sure I buy that gun violence events are driving searches for mental health. The t-tests are interesting, but cross-section time series are more appropriate. Looking at figure 1, I would argue that the increases in salience for mental health (thus increases in search volume) are occurring because mental health and issues surrounding it are becoming less stigmatized and more common in spaces where it simply was not discussed before. As a result, people may not have thought much about mental health issues before are now heading to their nearest computer and doing a quick Google search.

In general, I think this paper has something to add to the conversations on gun violence and mental health attitudes. However, at present, there is more work to be done on this paper so it meets a higher evidentiary standard.

Very minor note: I find the use of past tense, passive voice distracting. I know several discipline-specific associations (i.e., APA, APSA, and ASA) now recommend using present tense, active voice. I think that would also help this paper.

Reviewer #2: Dear authors,

I think you are rising an interesting and novel question. In my opinion you could make a further effort to improve and enrich your data analysis. As it is now, the analysis documents that the American society associates mental health and violence in the context of mass shootings. Additional effort could make it possible to draw conclusions of the type: mass shooting episodes increases societal association of violence and mental health.

1. You could exploit geographical variation. Media coverage and internet searches may be more intense in the state where the mass shooting took place. You could use state data to analyze whether searches of gun violence and mental health become more common in the state of the mass shooting right after the mass shooting with respect to other states. Technically, you could use something in the lines of a Difference-in-Difference estimation.

2. In your methodology, you compare searches before and after the event. In order to take into account pre-existing trends in your current data, you could use something in the lines of an Event Study methodology.

3. In the last section, you mention that the relationship between violence and mental health is a misconception. I think this constitutes an important justification for your research question. Hence, you could use it as part of your motivation in the introductory section.

6. PLOS authors have the option to publish the peer review history of their article (what does this mean?). If published, this will include your full peer review and any attached files.

Reviewer #1: No

Reviewer #2: No

---

## [Author Response · Author response to Decision Letter 0]

9 Jun 2020

Response to reviewers

 We are very grateful to the reviewers for providing insightful suggestions which we believe have helped us significantly improve the manuscript. Below are comments detailing edits made in this round of revisions. 

Editor 

We have edited the manuscript to meet style requirements. 

2. In your Methods section, please include additional information about your dataset and ensure that you have included a statement specifying whether the collection method complied with the terms and conditions for the websites from which you have collected data.

The methods were updated to provide more detail on the dataset:

“Google Trends provides search volume data by compiling a score which consists of relative frequency of searches of specific keywords at the time and location specified [1]. The data is publicly available (https://trends.google.com/trends/?geo=US). The content in this study is solely the responsibility of the authors and does not represent the official views of Google.” p. 6

Reviewer 1

1. There are several pieces of research that highlight the utility and validity of Google Trends missing from this piece. Some important ones are:

Mellon, Jonathan. 2013. “Where and When Can We Use Google Trends to Measure Issue Salience?” PS: Political Science & Politics 46 (2): 280–90. ‪https://doi.org/10.1017/S1049096513000279‬‬‬‬‬‬‬‬‬‬.

———. 2014. “Internet Search Data and Issue Salience: The Properties of Google Trends as a Measure of Issue Salience.” Journal of Elections, Public Opinion and Parties 24 (1): 45–72. ‪https://doi.org/10.1080/17457289.2013.846346‬‬‬‬‬‬‬‬‬‬.

Reilly, Shauna, Sean Richey, and J. Benjamin Taylor. 2012. “Using Google Search Data for State Politics Research An Empirical Validity Test Using Roll-Off Data.” State Politics & Policy Quarterly 12 (2): 146–59. ‪https://doi.org/10.1177/1532440012438889‬‬‬‬‬‬‬‬‬‬.

Thank you for the insightful suggestion. Firstly, the introduction has been edited to include these relevant investigations: 

“Capturing searches as they occur in real time rather than asking participants to recall their interest over time also removes confounds related to recall bias. Google Trends has been increasingly used to predict influenza epidemics [2], determine the effect of race on presidential polling [3], and even quantify interest in birth control and abortion [4]. Thus, it may lend itself particularly well to gathering public opinion and interest on topics that would otherwise be sensitive and difficult to gauge [5-7].” p. 5

2. A crucial take-away from these papers is that, when using Google Trends, the most appropriate model is a cross-sectional time series. In the case of this paper, the independent variable should be a fixed effect for the gun violence event which allows for a clean pre-post change estimation. As it stands now, this paper is descriptive—which is fine—but the use of Google Trends allows for so much deeper analysis. As for the mental health aspects, again, as this is merely descriptive with no large-scale content analysis for media agenda setting/framing/priming, I am not yet persuaded. I am just not sure I buy that gun violence events are driving searches for mental health. The t-tests are interesting, but cross-section time series are more appropriate. 

We agree that not accounting for each individual event as a fixed effect was a weakness of the prior version of the manuscript. We removed the analyses using t-tests to see whether search volumes changed before and after the events. In their place, we now use fixed effects panel regression using the Least Squares Dummy Variable (LSDV) approach [8] to examine searches before and after the events, accounting for each gun violence event as a fixed effect. An interaction of timepoint by whether the search occurred in the state where the gun violence event occurred was also analyzed, in line with reviewer 2 point 1’s suggestion. We also used the fixed effects panel regression LSDV approach to analyze whether related searches of interest for our key words increased in proportion after the gun violence events. The methods and results have been updated accordingly: 

“2.3c. Main effect analyses. Search terms were extracted for the week before the 15 events, and the week after the 15 events, for each of the 50 states and Washington D.C. The data extraction took place in July 2019. Before and after scores of relative search volume were extracted for key terms (“Mental Health” and “Gun(s)”). A fixed effects panel regression using the Least Squares Dummy Variable (LSDV) approach was then conducted [8]. Searches from each state the week before and after the events were compared, with the 15 events accounted for as fixed effects. An interaction of state where the event occurred by timepoint was tested to see whether change in search volumes in the state the event occurred would be significantly different from change in search volumes in states where the event did not occur.” p. 9

“2.3e. Related search terms for the week before and week after. Related searches for “Gun(s)” and “Mental Health” key words were extracted for the week before and the week after the 15 events. The week before was chosen as a comparison point prior to the event taking place. Each related search was dummy coded as “topic of interest” or “not topic of interest.” Percentage of topics of interest among total related searches were computed for the week before and after (Figs 3,4). A fixed effects panel regression using the LSDV approach (to account for events as fixed effects) was conducted to determine whether proportion of related search terms would increase after the event occurred.” p. 10

“3.3. Keyword searches before and after mass shooting events. For the “mental health” key term, the model was significant, R = 0.30, R2 = 0.09, F = 0.892, p <0.001. Relative to the week before the event, there was a trend toward larger volume of “mental health” searches, b = 1.47, SE = 0.82, t = 1.80, p = 0.07. A main effect was not observed for whether the search occurred in the state the event occurred, b = -13.25, SE = 9.20, t = -1.44, p = 0.15. The interaction of event timepoint and search occurring in-state/out-of-state was not significant, b = 0.80, SE = 5.82, t = 0.14, p = 0.89. 

 The model of searches for the “gun” key term was likewise significant, R = 0.35, R2 = 0.12, F = 12.04, p <0.001. Relative to the week before the event, a larger volume of “gun” searches was observed the week after, b = 5.42, SE = 0.64, t = 8.45, p < 0.001. A significant main effect was not observed for whether the search occurred in the state the event occurred (b = -10.96, SE = 7.26, t = -1.51, p = 0.13), and neither was an interaction (b = 6.97, SE = 4.59, t = 1.52, p = 0.13). As expected, the model for searches for the “guns” key term was also significant, R = 0.33, R2 = 0.11, F = 10.70, p <0.001. Compared to the week before events occurred, a larger volume of “guns” searches was observed the following week, b = 4.80, SE = 0.79, t = 6.05, p < 0.001. As was the case for other key terms, a significant main effect was not observed for the search occurring in the state where the event occurred (b = -15.44, SE = 8.95, t = -1.72, p = 0.09), nor was a significant interaction observed (b = 7.07, SE = 5.66, t = 1.25, p = 0.21).” p. 11

“The model of related searches before and after the event for the “mental health” key term was significant, R = 0.25, R2 = 0.06, F = 4.04, p <0.001--related searches that pertained to law, politics/government, crime/violence, mass shootings, stigma, and guns significantly increased in proportion the week after the event, β = 0.08, t = 2.44, p = 0.015. Likewise, the model of related searches for “gun(s)” key terms was significant, R = 0.28, R2 = 0.08, F = 12.39, p <0.001. The “gun(s)” related searches that pertained to law, politics/government, crime/violence, mass shootings, stigma, and mental health significantly increased in proportion the week after the event, β = 0.22, t = 10.35, p < 0.001.” p. 13

“Significant differences in search volume were not found in the states where the shooting events occurred (relative to states in which the event did not occur); future investigations could instill more fine-grained approaches to inform the manners in which geography could interact with search volumes.” p. 16

3. Looking at figure 1, I would argue that the increases in salience for mental health (thus increases in search volume) are occurring because mental health and issues surrounding it are becoming less stigmatized and more common in spaces where it simply was not discussed before. As a result, people may not have thought much about mental health issues before are now heading to their nearest computer and doing a quick Google search. In general, I think this paper has something to add to the conversations on gun violence and mental health attitudes. However, at present, there is more work to be done on this paper so it meets a higher evidentiary standard.

The Reviewer offers a viable alternative explanation to the interpretation of our findings. The Reviewer suggests that it is possible that more searches are a result of less societal stigma against mental health and more open dialogs, and it is these conversations that are leading to increased searches. Although likely accounting for some of the findings, we feel that it is far more likely that our findings reflect response to mass shootings. Using the monthly data of search volumes for “mental health” over time, we ran analyses to see whether there would be an effect of shooting events while accounting for the effect of searches increasing over time. As expected and noted by the reviewer, there was an effect of time such that later months evidenced a larger volume of “mental health” searches, β = 0.69, t = 10.01, p<0.001. However, we also observed a significant interaction of time by event, β = 0.85, t = 4.17, p<0.001, such that accounting for the general increase in searches overtime, the observed increase in “mental health” searches was steeper during months in which shooting events occurred. These analyses suggest that mass shooting events could contribute to “mental health” search volumes, even when accounting for the general increase in “mental health” searches over time. Further, analyses presented in Reviewer 1 point 2 with regards to key term related searches show that “mental health” searches that pertained to law, politics/government, crime/violence, mass shootings, stigma, and guns significantly increased in proportion the week after shooting events. We would not have expected topics of related searches to increase in this manner if the effect of increased searches the week after events was accounted for by a general overall increase in mental health interest over the years (rather, we would have expected related searches pertaining these topics to remain comparable in proportion the weeks before and after the events). Nonetheless, we agree that the current investigation is preliminary, and many future studies will be needed to come to any definitive conclusions. To further address this point, we have also added to our Limitations section acknowledging that current results are preliminary, and further evidence is needed to strengthen the line of inquiry:

“Although our preliminary results suggest that greater numbers of searches after mass shootings, it is possible that in some circles inroads have been made to reduce stigma around mental health and that stigma reduction efforts have contributed to increased Google searching in an effort for people to learn more. More work at the individual level to understand what drives these searches would add an additional level of context and depth to this field of inquiry.” p. 18

4. Very minor note: I find the use of past tense, passive voice distracting. I know several discipline-specific associations (i.e., APA, APSA, and ASA) now recommend using present tense, active voice. I think that would also help this paper.

 The manuscript underwent a critical read-through, and has been edited throughout to increase usage of active voice. 

Reviewer 2

1. You could exploit geographical variation. Media coverage and internet searches may be more intense in the state where the mass shooting took place. You could use state data to analyze whether searches of gun violence and mental health become more common in the state of the mass shooting right after the mass shooting with respect to other states. Technically, you could use something in the lines of a Difference-in-Difference estimation.

We agree that this is an interesting and relevant question that we very much appreciated. As detailed in Reviewer 1 point 2, an interaction analysis was added to the new version of the manuscript. Although we agree that this was an idea worthy of pursuing, findings did not suggest a significant interaction of timepoint and the search occurring in-state/out of state. 

2. In your methodology, you compare searches before and after the event. In order to take into account pre-existing trends in your current data, you could use something in the lines of an Event Study methodology.

 This is indeed a crucial consideration, and new analyses are detailed in Reviewer 1 points 2 and 3. 

3. In the last section, you mention that the relationship between violence and mental health is a misconception. I think this constitutes an important justification for your research question. Hence, you could use it as part of your motivation in the introductory section.

 We appreciate the Reviewer’s point and have edited the Introduction to highlight that the magnitude of the relation between mental health and violence is much smaller than is typically portrayed in the media. As a result of this point of emphasis, the Introduction now better motivates the rationale behind the study.

“Strikingly, there is strong evidence that media portrayals of topics related to mental health are likely to have an impact on individual attitudes toward mental illness [9-16]. For example, studies have found that increased depictions of untreated individuals diagnosed with mental illness can result in increased social distance from groups suffering from mental illness; this, in turn, has been found to lead to increased isolation of these individuals [14, 17-19]. Furthermore, stigmatizing attitudes toward mental illness have been linked to reduced support for policy supporting mental health [9].Understanding how public perception is impacted on an individual basis could thus be critical for gauging the societal effects of media coverage of mental health concerns. In the case of violence, it could be particularly impactful, given evidence that the link between violence and mental illness is reliable but often overestimated in magnitude [20], with population-wide studies having shown that only around 4% of violence in the U.S. can be attributed to individuals diagnosed with a psychiatric disorder [21].” p. 4

References

1. Takeda F, Wakao T. Google search intensity and its relationship with returns and trading volume of Japanese stocks. Pacific-Basin Finance Journal. 2014;27:1-18.

2. Ginsberg J, Mohebbi MH, Patel RS, Brammer L, Smolinski MS, Brilliant L. Detecting influenza epidemics using search engine query data. Nature. 2009;457(7232):1012.

3. Stephens-Davidowitz S. The cost of racial animus on a black candidate: Evidence using Google search data. Journal of Public Economics. 2014;118:26-40.

4. Kearney MS, Levine PB. Media influences on social outcomes: The impact of MTV's 16 and pregnant on teen childbearing. American Economic Review. 2015;105(12):3597-632.

5. Mellon J. Where and when can we use Google Trends to measure issue salience? PS: Political Science & Politics. 2013;46(2):280-90.

6. Mellon J. Internet search data and issue salience: The properties of Google Trends as a measure of issue salience. Journal of Elections, Public Opinion & Parties. 2014;24(1):45-72.

7. Reilly S, Richey S, Taylor JB. Using Google search data for state politics research: An empirical validity test using roll-off data. State Politics & Policy Quarterly. 2012;12(2):146-59.

8. Allison PD. Fixed effects regression models: SAGE publications; 2009.

9. Barry CL, McGinty EE. Stigma and public support for parity and government spending on mental health: a 2013 national opinion survey. Psychiatric Services. 2014;65(10):1265-8.

10. Barry CL, McGinty EE, Vernick JS, Webster DW. After Newtown—public opinion on gun policy and mental illness. New England journal of medicine. 2013;368(12):1077-81.

11. McGinty EE, Webster DW, Barry CL. Effects of news media messages about mass shootings on attitudes toward persons with serious mental illness and public support for gun control policies. American Journal of Psychiatry. 2013;170(5):494-501.

12. Stuart H. Media portrayal of mental illness and its treatments. CNS drugs. 2006;20(2):99-106.

13. Angermeyer MC, Matschinger H. The effect of violent attacks by schizophrenic persons on the attitude of the public towards the mentally ill. Social Science & Medicine. 1996;43(12):1721-8.

14. Martin JK, Pescosolido BA, Tuch SA. Of fear and loathing: The role of'disturbing behavior,'labels, and causal attributions in shaping public attitudes toward people with mental illness. Journal of health and social behavior. 2000:208-23.

15. Sieff E. Media frames of mental illnesses: The potential impact of negative frames. Journal of Mental Health. 2003;12(3):259-69.

16. Thornton JA, Wahl OF. Impact of a newspaper article on attitudes toward mental illness. Journal of community psychology. 1996;24(1):17-25.

17. McGinty EE, Goldman HH, Pescosolido B, Barry CL. Portraying mental illness and drug addiction as treatable health conditions: effects of a randomized experiment on stigma and discrimination. Social Science & Medicine. 2015;126:73-85.

18. Lauber C, Nordt C, Falcato L, Rössler W. Factors influencing social distance toward people with mental illness. Community mental health journal. 2004;40(3):265-74.

19. Corrigan PW, Edwards AB, Green A, Diwan SL, Penn DL. Prejudice, social distance, and familiarity with mental illness. Schizophrenia bulletin. 2001;27(2):219-25.

20. Glied S, Frank RG. Mental illness and violence: Lessons from the evidence. American journal of public health. 2014;104(2):e5-e6.

21. Fazel S, Grann M. The population impact of severe mental illness on violent crime. American journal of psychiatry. 2006;163(8):1397-403.

---

## [Decision Letter · Decision Letter 1]

1 Jul 2020

Using search engine data to gauge public interest in mental health, politics and violence in the context of mass shootings

PONE-D-20-02370R1

Dear Dr. Vargas,

We are pleased to inform you that your manuscript has been judged scientifically suitable for publication and will be formally accepted for publication once it meets all outstanding technical requirements.

Kind regards,

Luis M. Miller, Ph.D.

Academic Editor

PLOS ONE

Reviewers' comments:

Reviewer's Responses to Questions

**Comments to the Author**

1. If the authors have adequately addressed your comments raised in a previous round of review and you feel that this manuscript is now acceptable for publication, you may indicate that here to bypass the “Comments to the Author” section, enter your conflict of interest statement in the “Confidential to Editor” section, and submit your "Accept" recommendation.

Reviewer #1: All comments have been addressed

Reviewer #2: All comments have been addressed

2. Is the manuscript technically sound, and do the data support the conclusions?

Reviewer #1: Yes

Reviewer #2: Yes

3. Has the statistical analysis been performed appropriately and rigorously? 

Reviewer #1: Yes

Reviewer #2: Yes

4. Have the authors made all data underlying the findings in their manuscript fully available?

Reviewer #1: Yes

Reviewer #2: Yes

5. Is the manuscript presented in an intelligible fashion and written in standard English?

Reviewer #1: Yes

Reviewer #2: Yes

6. Review Comments to the Author

Reviewer #1: All of my concerns have been addressed. The authors explain their edits and thinking very clearly. I am persuaded by their arguments and think this paper is a contribution to several literatures.

Reviewer #2: I am overall satisfied with the authors' response to my comments. I just have one suggestion: it would be great if you wrote the equation you estimate in the paper.

7. PLOS authors have the option to publish the peer review history of their article (what does this mean?). If published, this will include your full peer review and any attached files.

Reviewer #1: No

Reviewer #2: No